# Positive Mental Health Questionnaire-Short Form (PMHQ-SF18): Psychometric Properties of the Spanish Version

**DOI:** 10.3390/nursrep15110392

**Published:** 2025-11-07

**Authors:** Maria Teresa Lluch-Canut, Montserrat Puig-Llobet, Maria Aurelia Sánchez-Ortega, Carmen Moreno-Arroyo, Antonio R. Moreno-Poyato, Juan F. Roldán-Merino, Miguel Ángel Hidalgo-Blanco, Carmen Ferre-Grau, Núria Albacar-Riobóo, Carlos Sequeira, Sara Sanchez-Balcells, Susana Mantas-Jiménez, Marta Prats-Arimon, Zaida Agüera

**Affiliations:** 1Departament d’Infermeria de Salut Pública, Salut Mental i Maternoinfantil, Facultat d’Infermeria, Universitat de Barcelona, L’Hospitalet de Llobregat, 08907 Barcelona, Spainsara.sanchez@ub.edu (S.S.-B.);; 2Research Group in Mental Health, Psychosocial and Complex Nursing Care (NURSEARCH), Facultat d’Infermeria, Universitat de Barcelona, L’Hospitalet de Llobregat, 08007 Barcelona, Spain; juan.roldan@sjd.edu.es (J.F.R.-M.);; 3Departament de Fonaments Professionals i Gestió en Salut, Facultat de Ciències de la Salut i el Benestar, Universitat de Vic, 08007 Barcelona, Spain; 4Departament d’Infermeria Fonamental i Clínica, Facultat d’Infermeria, Universitat de Barcelona, L’Hospitalet de Llobregat, 08007 Barcelona, Spain; miguelhidalgo@ub.edu; 5Department of Nursing Research Group (GRIN), Bellvitge Biomedical Research Institute (IDIBELL), L’Hospitalet de Llobregat, 08007 Barcelona, Spain; 6Campus Docent Sant Joan de Déu-Fundació Privada, School of Nursing, Sant Boi de Llobregat, 08007 Barcelona, Spain; 7Department of Nursing, Faculty of Nursing, University of Rovira i Virgili, 43003 Tarragona, Spain; 8Center for Health Technology and Services Research-Health Research Network from de Lab to the Community CINTESIS@RISE/ESEP, Nursing School of Porto, 4169-005 Porto, Portugal; carlossequeira@esenf.pt; 9Departament d’Infermeria, Facultat d’Infermeria, Universitat de Girona, 17003 Girona, Spain; susana.mantas@udg.edu; 10Health and Health Care Research Group, Universitat de Girona, 17004 Girona, Spain; 11Psychoneurobiology of Eating and Addictive Behaviors Group, Neurosciences Programme, Bellvitge Biomedical Research Institute (IDIBELL), L’Hospitalet de Llobregat, 08908 Barcelona, Spain; 12CIBER Fisiopatología Obesidad y Nutrición (CIBERobn), Instituto de Salud Carlos III, 28029 Madrid, Spain

**Keywords:** mental health screening, mental health promotion, positive mental health, psychological well-being, psychometric properties, factor analysis

## Abstract

**Background**: The construct of positive mental health (PMH) is defined as the basis of individuals’ psychological well-being and their ability to function effectively and cope with life’s challenges. The Positive Mental Health Questionnaire (PMHQ) is a reliable tool for assessing the PMH factors, but its length (39 items) can pose challenges in certain contexts and populations. This highlights the need for an abridged version of the questionnaire that requires less time to administer. Therefore, the main aim was to validate the Spanish 18 item-shortened version of the PMHQ (PMHQ-SF18). **Methods**: The sample consisted of 574 nursing students. Psychometric analyses were carried out based on construct validity, criterion validity, and internal consistency. A confirmatory factor analysis was conducted to ascertain whether the internal structure was consistent with the model of the previously validated Portuguese brief version. **Results**: The results supported the good psychometric properties of the instrument, with adequate validity and reliability. Confirmatory factor analysis revealed optimal goodness-of-fit values, supporting the six-factor structure. The overall Cronbach’s alpha was 0.83. **Conclusions**: The findings suggest that the PMHQ-SF18 is a valid and reliable instrument, comparable to the original version, but with the added benefits of being shorter, quicker, and easier to administer. Consequently, it may be particularly useful for population-based screening studies and for monitoring change following positive mental health promotion interventions. Its abridged format makes it particularly suitable for assessing individuals with specific characteristics or in contexts where time is limited, and more concise instruments are required, for example, in primary care or critical care.

## 1. Introduction

The promotion of mental health is one of the main health challenges of recent decades. Already in 2004, the World Health Organization (WHO) placed special emphasis on the need to address positive mental health (PMH). This construct of PMH was defined by the WHO as the basis of well-being and effective functioning of individuals, always consistent with its interpretation across cultures [1]. Therefore, promoting PMH is an inherent element of public health because mental well-being directly influences overall health, social functioning, and quality of life.

The concept of PMH was first introduced by Marie Jahoda [2], who defined it as a sense of physical, psychological, and social well-being, which implies three domains for the individual: perception of self-actualization, control over one’s environment, and autonomy. Afterwards, Lluch-Canut [3] defined the PMH construct as “*a dynamic and fluctuating state in which a person tries to feel and be the best possible within his or her circumstances*.” Lluch-Canut’s multifactorial model of PMH diverges from the traditional psychopathology-based approach to mental health. Rather than emphasizing the diagnosis and treatment of mental disorders, this model focuses on promoting protective factors that support well-being and resilience. This model involves six main interrelated factors: personal satisfaction (F1), prosocial attitude (F2), self-control (F3), autonomy (F4), problem solving and self-actualization (F5), and interpersonal relationship skills (F6).

The evidence for promoting PMH depends on its correct definition, assessment, measurement, and recording [1]. Thus, it is crucial to have robust psychometric tools developed under an evidence-based conceptual framework that allows us to accurately assess PMH and its dimensions. In accordance with the multifactorial model proposed by Lluch-Canut [3], this same author developed the Positive Mental Health Questionnaire (PMHQ) to operationalize the model and provide a useful instrument [3]. The PMHQ consists of 39 items distributed among the 6 factors that define the construct. Previous studies have been carried out to validate of this questionnaire among Spanish students at four university nursing schools in Catalonia, demonstrating good psychometric properties of the instrument, both in the exploratory factor analysis [4] and the confirmatory factor analysis [5].

The PMHQ has proven to be a useful, valid, and reliable tool for assessing PMH in different populations, both clinical [6] and non-clinical [7,8]. It has been adapted and validated in different countries and languages [9,10]. In addition, the PMHQ has been shown to be effective in assessing the effects or changes resulting from psychosocial nursing interventions aimed at improving PMH in different populations, including: undergraduate nursing students [11], pregnant women [12], adults with chronic diseases, both mental health disorders [13] and physical health conditions [14], or even in informal caregivers of patients with chronic diseases [15,16,17,18]. However, in certain settings its administration may be limited by lack of time or by the specific characteristics of the users (e.g., intensive care units, primary care). This highlights the need for a shortened version of the instrument, which is just as valid and reliable, but requires less time to administer.

In the literature, we can find some abbreviated scales to assess PMH, such as the 19-item scale called PMH-19 [19]; the Mental Health Continuum-Short Form (MHC-SF), which measures feelings of well-being through 14 items [20,21]; the unidimensional PMH Scale, which consists of only nine items [22]; and the Rapid Positive Mental Health (RPM) instrument, which is a six-item unidimensional scale [23]. However, none of these brief scales are based on Lluch’s multifactorial model of PMH nor do any of them include the six proposed dimensions.

Based on the multifactorial model of PMH, Sequeira et al. [24] have recently developed a validation in Portuguese of an abridged version of the PMHQ with 18 items (i.e., PMHQ-SF18). The PMHQ-SF18 Portuguese version retains the six original dimensions and has adequate psychometric properties. However, to our knowledge, no validation in a Spanish population is available so far. Therefore, the main objectives of this study were to analyze the psychometric properties of the Spanish version of the PMHQ-SF18, and to examine its association with sociodemographic data in a sample of undergraduate nursing students. In this context, the research question guiding this study was: *Does the Spanish version of the PMHQ-SF18 demonstrate adequate psychometric properties and maintain the original six-factor structure when applied to Spanish nursing students?*

## 2. Methods

### 2.1. Design

This validation study adopted a cross-sectional design within a post-positivist framework, guided by a quantitative approach and aligned with STROBE reporting guidelines.

### 2.2. Setting and Participants

The sample consisted of 574 undergraduate students from the Faculty of Nursing at the University of Barcelona. Following the guidelines proposed to correctly apply factorial techniques [25], we estimated a minimum sample size of 90 students. In addition, we adhered to the recommendations of [26], who advise a sample size of 300–500 participants for a successful performance of factorial techniques.

All participants were recruited and assessed during the first two weeks of their undergraduate course, during the academic years 2021–2022 and 2022–2023. Inclusion criteria included voluntary participation in this study with the signing of informed consent, and fulfillment of all questionnaire items.

### 2.3. Variables and Instruments

Sociodemographic data were obtained by means of an ad hoc questionnaire, including variables such as gender, age, nationality, marital status, employment, people with whom participants lived, and assessment of such cohabitation.

*Positive Mental Health Questionnaire short version* (PMHQ-SF18). Based on the 39-item PMH questionnaire, a shortened version of 18 items was used, which was validated in Portuguese [24]. This questionnaire is composed of 18 items distributed across the 6 factors, with 3 items per factor, that define the construct: F1-personal satisfaction, F2-prosocial attitude, F3-self-control, F4-autonomy, F5-problem solving and self-actualization, and F6-interpersonal relationship skills. There are direct and inverse items that are answered using a Likert scale ranging from 1 to 4 (being 1 = *always or almost always*, 2 = *quite often*, 3 = *sometimes*, and 4 = *rarely or never*). Scores are obtained for each factor as well as for the total score ranging from 18 to 72. Higher scores indicate better levels of PMH. The Portuguese brief version shows good psychometric properties, with only Factor 2 presenting a lower Cronbach’s alpha (0.60) [24].

### 2.4. Translation Procedure

Although back-translation is a widely recommended step in cross-cultural adaptation to ensure semantic and conceptual equivalence, this procedure was not applicable in the present study. The brief version of the Positive Mental Health Questionnaire (PMHQ-SF18) was derived directly from the original Spanish version, rather than from a translated version. Therefore, the adaptation process involved item selection and psychometric validation within the same linguistic and cultural context, eliminating the need for back-translation.

### 2.5. Procedures

In the second week after the beginning of the course, the sociodemographic questionnaire and the PMHQ were administered to all students enrolled in the Psychosocial Sciences Applied to Health subject in the first year of the Nursing degree course. Beforehand, everyone was informed that participation in this study was for research purposes only, was completely anonymous, voluntary, and had no impact on the academic course. Only those who voluntarily agreed to participate and signed an informed consent form were included in this study.

### 2.6. Statistical Analysis

Descriptive analyses were carried out using measures of central tendency (means, medians) and dispersion [standard deviation (SD), variance and first and third quartiles] for quantitative variables, and frequencies and percentages for qualitative ones. Acceptability was estimated based on score distribution: missing data (<5%) and floor and ceiling effects <15% (i.e., the percentage of individuals scoring the minimum and maximum scores, respectively) [27].

Reliability of the instrument was established using Cronbach’s alpha to demonstrate internal consistency (i.e., Cronbach alpha values of 0.70 or higher indicate acceptable internal consistency) [27]. We also determined the internal consistency using the corrected item-total correlation (acceptable limit ≥ 0.2). The polychoric correlation of each item with its corresponding factor was also calculated.

Spearman’s correlation coefficient was used to test criterion (convergent and discriminant) validity. Convergent validity refers to the degree to which each factor is related to the other factors and to the overall PMH total score. Discriminant validity refers to the degree to which each factor differs from the other factors of the same construct. Therefore, robust construct validity would be supported if there were strong correlations between each factor and the PMH total score, while correlations between factors remained low, indicating that they measure different aspects of the construct [5].

Construct validity was tested through confirmatory factor analysis (CFA). Goodness of fit was evaluated using the Comparative Fit Index (CFI; minimum criterion ≥ 0.90), the Normed Fit Index (NFI; minimum criterion ≥ 0.90), and the root mean squared error of approximation (RMSEA; minimum criterion ≤ 0.06) [28]. Standardized factor loadings were used to determine the items that belong to each latent variable. A loading of 0.40 or lower is considered deficient [29] and removal of this item may be considered.

The analysis of the relationship between the sociodemographic variables and the PMHQ-SF18 factors and total score was performed using Student’s *t*-test or ANOVA, according to the application criteria. Statistical significance was set at a *p*-value of 0.05. The effect size was estimated through the standardized Cohen’s-d coefficient, considering small effect as |d| > 0.20, a mild-moderate as |d| > 0.50, and large-high as |d| > 0.80.

Data analysis was conducted with SPSS v.27 and R v.4.1.0 statistical software for Windows.

## 3. Results

### 3.1. Sociodemographic Data

Most of the participants were female, single and living with their parents. The socio-demographic data of the participants are summarized in Table 1.

### 3.2. Item Analysis

Table 2 presents the item analyses. The mean item values ranged from 2.65 (item 4) to 3.74 (item 2), and the SD ranged from 0.43 (item 14) to 0.95 (item 16). There were no missing data. The floor effect was negligible for all items. The ceiling effect was above the maximum value established (>15%) in all the items. Polychoric correlations item-total factor are also shown in Table 2.

### 3.3. Reliability

Internal consistency was high for the total score of PMH-SF18 (α = 0.83). Cronbach’s α scores for each factor were as follows: personal satisfaction (F1) = 0.76, prosocial attitude (F2) = 0.45, self-control (F3) = 0.79, autonomy (F4) = 0.81, problem solving and self-actualization (F5) = 0.64, and interpersonal relationship skills (F6) = 0.69.

Corrected item-total correlations were adequate (>0.20) for most of the items (16 of the 18 items), indicating that virtually all items contributed to the internal consistency of the questionnaire. Only two items showed corrected item-total correlation coefficients below 0.20 (items 8 and 14). Interestingly, both items belong to F2-prosocial attitude (Table 2).

### 3.4. Criterion (Convergent and Discriminant) Validity

Table 3 shows the Spearman correlation coefficients. Discriminant validity was confirmed with the inter-factor correlation coefficients ranging from very weak to weak (r = 0.07 to r = 0.55) indicating that most factors measured distinct constructs. This finding supports the discriminant validity of each factor. The least weak correlations were found between F1 (personal satisfaction) and F4 (autonomy) (rho = 0.55) and between F3 (self-control) and F5 (problem solving and self-fulfillment) (rho = 0.51). The weakest inter-factor correlation was between F1 (personal satisfaction) and F2 (pro-social attitude) (rho = 0.07).

In terms of convergent validity, the strongest correlations were found between the PMH total score and most of the dimensions. The F5 (problem solving and self-actualization) was the most highly correlated with the PMH total score (rho = 0.80), while F2 (prosocial attitude) showed the weakest correlation (rho = 0.37).

### 3.5. Construct Validity: Confirmatory Factor Analysis

Construct validity was also tested through CFA to assess the relationship defined by the instrument between the items and the factors. All the goodness-of-fit indexes of the CFA reached the critical limits established: CFI = 0.99 (minimum criterion ≥ 0.9), NFI = 0.98 (minimum criterion ≥ 0.9), and RMSEA [95%CI] = 0.0462 [0.0391; 0.0533] (minimum criterion ≤ 0.06). According to the results, the model proposed for the factors fit the data satisfactorily. In addition, no item had a factor loading of less than 0.40, so none was considered for removal [29]. Item four was the one that contributed the most (0.88). All factors were positively related. The most highly related were F4 (autonomy) and F1 (personal satisfaction) (0.69), followed by F5 (problem solving) and F2 (prosocial attitude) (0.63). The least related were F2 (prosocial Attitude) and F1 (personal Satisfaction) (0.12) (see Figure 1).

### 3.6. Association Between PMH Factors and Sociodemographic in Known Groups

No significant differences were found in most factors with respect to gender. The only significant difference was in self-control, where men scored significantly higher. The effect size indicated a medium effect size. Table 4 shows the comparison between known groups by gender.

Concerning the living situation, those students who lived with a partner (in an intimate relationship) reported statistically higher levels of personal satisfaction, self-control, autonomy, and interpersonal relationships skills, as well as higher levels of overall PMH, than those who lived with their parents. In addition, participants living with a partner showed higher perceived self-control, autonomy, and total PMH than those living with roommates or friends or others. All the significant comparisons showed an effect size ranging from mild-moderate (|d| > 0.50) to large (|d| > 0.80). Table 5 shows the comparison between known groups by coexistence.

## 4. Discussion

The current study aimed to validate the Spanish version of the PMHQ-SF18. Overall, our results confirm that this brief version of the PMHQ retains the well-known, robust psychometric properties of the original questionnaire, but with the advantage of being shorter, in line with the brief version previously validated in Portuguese [24].

The analysis of the items detected no floor effects. However, it should be noted that all the items showed high ceiling effects. Although this detracts from the discriminative power of the items, these results are in line with those reported in previous studies conducted with the original PMHQ [5]. Similar to previous studies, the highest ceiling effects (>75%) were observed for two of the three items that comprise Factor 2 (Prosocial attitude). Specifically, the two items are: item 2: ‘*I feel that I am someone who can be trusted*’ and item 14 ‘*I like to help others*.’ This might be because all participants were nursing students and therefore, people characterized by a clear vocation to help and care for others.

The PMHQ-SF18 obtained a good level of reliability in terms of internal consistency for the total score. Three of the six proposed factors (F1, F3, and F4) obtained good levels of reliability (Cronbach’s alpha > 0.70). However, three factors such as prosocial attitude (F2), problem-solving and self-actualization (F5), and interpersonal relationship skills (F6) scored below 0.70, with F2 having the lowest Cronbach’s alpha coefficient. Moreover, this was not an unexpected finding because it had previously been found in other studies using the full 39-item PMHQ, both in studies conducted with clinical samples [6] and in those conducted with university students [5,30], where the Cronbach’s alpha value of F2 did not exceed 0.60. This is also consistent with the values reported in the Portuguese validation of this abbreviated scale [24]. Low alpha values for F2 may be attributed to psychometric factors such as the reduced number of items or item wording, which should be considered in future research.

The results also provide evidence of the criterion validity of the instrument. Accordingly, discriminant validity confirmed that the six subscales have good levels of discrimination and each factor measures a distinct aspect of the PMH model. In addition, convergent validity confirmed that nearly all factors correlated with the PMH total score. This suggests that almost all factors are adequate for explain the PMH construct. Although F2 should be further reviewed, these results are consistent with those provided by the original large instrument [5].

In addition, the results of the CFA support the multifactorial model of PMH that underpins the questionnaire. Despite the reduction in the number of items, the data support that the 18 items are adequately distributed in the same six factors as the extended PMHQ (3 items per factor), namely: personal satisfaction (F1), prosocial attitude (F2), self-control (F3), autonomy (F4), problem-solving and self-actualization (F5), and interpersonal relationship skills (F6). These results are similar to those obtained in the Portuguese version of Sequeira et al. [24] in which the factor loadings of the items ranged from 0.41 to 0.85.

Differences in some sociodemographic variables were revealed in the sample studied. In terms of gender, differences were only observed in factor F3 (self-control), where male participants reported greater control of their own emotions and negative thoughts in situations of conflict or stress than female participants. These data are consistent with those reported in previous studies, indicating that female college students tend to have higher levels of perceived stress and use emotion-focused, maladaptive coping mechanisms more often than male students [31,32]. However, they are not in line with a recent study of Portuguese university students in which male undergraduates reported higher personal satisfaction and autonomy compared to their female counterparts, but not significantly higher self-control [7]. This highlights potential cultural differences that should be addressed in future cross-cultural studies to better understand PMH in young people [33].

It is also important to consider that the participants were first-year nursing students from universities in Catalonia, a region with its own cultural and linguistic characteristics within Spain. Cultural context may influence how individuals interpret and respond to items related to mental health constructs, potentially affecting factor structure and reliability indices.

Regarding the living situation, our results indicate that university students who live with a partner have a higher perception of PMH (i.e., greater personal satisfaction, self-control, autonomy, and interpersonal skills) than those who live with their parents or in other cohabitation situations. These results are consistent with previous studies, suggesting that living with a partner might be a protective factor in young people against psychological distress and mental health issues [34]. This finding aligns with research showing strong links between psychological well-being, resilience, and positive mental health in university students, which became even more critical after the COVID-19 pandemic due to increased stress and isolation [35]. Although interventions such as mindfulness and emotional intelligence training can improve outcomes, their effectiveness varies, highlighting the need for comprehensive strategies to foster resilience and well-being in this population [36].

The validation of the PMHQ-SF18 offers significant implications for mental health nursing practice, education, and research. As a concise and psychometrically robust instrument, the PMHQ-SF18 facilitates the efficient assessment of positive mental health (PMH) among nursing students and potentially in broader populations. Its brevity makes it particularly suitable for use in clinical settings where time constraints are common, such as in primary care, emergency departments, and critical care units.

For mental health nurses, the availability of a reliable and valid compact tool enhances the capacity to screen for psychological well-being, monitor changes over time, and evaluate the impact of mental health promotion interventions. This aligns with the nursing role in fostering holistic care, which includes promoting resilience, coping strategies, and emotional well-being.

Moreover, the PMHQ-SF18 can be integrated into nursing education to support the development of self-awareness and mental health literacy among students, contributing to their personal and professional growth. In research contexts, it enables large-scale data collection with minimal burden, supporting evidence-based practices and policy development in mental health nursing.

### 4.1. Limitations and Strengths

The present study has some limitations. Firstly, validation was carried out only with first-year nursing students from a single university. Thus, considering that the sample is specific and homogeneous, we should be cautious in generalizing the results. In addition, the sample was predominantly female, reflecting the demographic reality of nursing programs at the participating institutions. This imbalance may have influenced the findings and further limits the generalizability of the results. However, given that previous psychometric analyses of both the original 39-item PMHQ and the abbreviated Portuguese version of the PMHQ-SF18 were conducted with university students, we consider it appropriate that the validation of this abridged Spanish version also be carried out with a comparable sample. Secondly, the stability of the questionnaire was not assessed in the present study. Therefore, future studies should analyze test–retest correlations to examine the suitability of the instrument for evaluating the effectiveness of the intervention programmes. In addition, future research should explore the performance of the scale in more diverse Spanish-speaking populations and compare the results across different cultural settings to strengthen generalizability. Such efforts would help determine whether the observed patterns are consistent across genders and cultural contexts, thereby improving the external validity of the PMHQ-SF18. Furthermore, data on health and mental well-being were not collected, which could have enriched the interpretation of the results and offered a more comprehensive understanding of the phenomenon. Finally, sexual identity was not assessed, which could have provided a valuable dimension to the analysis, as relationship dynamics may vary significantly depending on this variable. This aspect will be considered in future research.

### 4.2. Linking Evidence to Action

Robust instruments are needed for the assessment of positive mental health, as well as for the monitoring of changes following mental health nursing interventions. In this regard, the original 39-item Positive Mental Health Questionnaire had already demonstrated its effectiveness for clinical nursing practice. Therefore, having an instrument as valid and reliable as the original, but shorter (18 items) and quicker to administer, may be useful for population-based screening studies, as well as for the assessment of individuals with specific characteristics or in settings where time is limited and concise instruments are required, for instance, in primary care or critical care.

## 5. Conclusions

In conclusion, the Spanish version of the 18-item brief questionnaire showed good psychometric properties. The results provide evidence that the PMHQ-SF18 is a valid and reliable instrument for assessing PMH. Importantly, it preserves the integrity of the six-factor multifactorial model of PMH. Given its brevity and ease of use, the PMHQ-SF18 stands out as a highly effective tool for use in settings where time is limited or where the specific characteristics of the target population demand a more concise assessment. Its potential for broad application makes it a valuable instrument for both research and clinical practice, especially in large-scale screening and mental health promotion initiatives.

## Figures and Tables

**Figure 1 nursrep-15-00392-f001:**
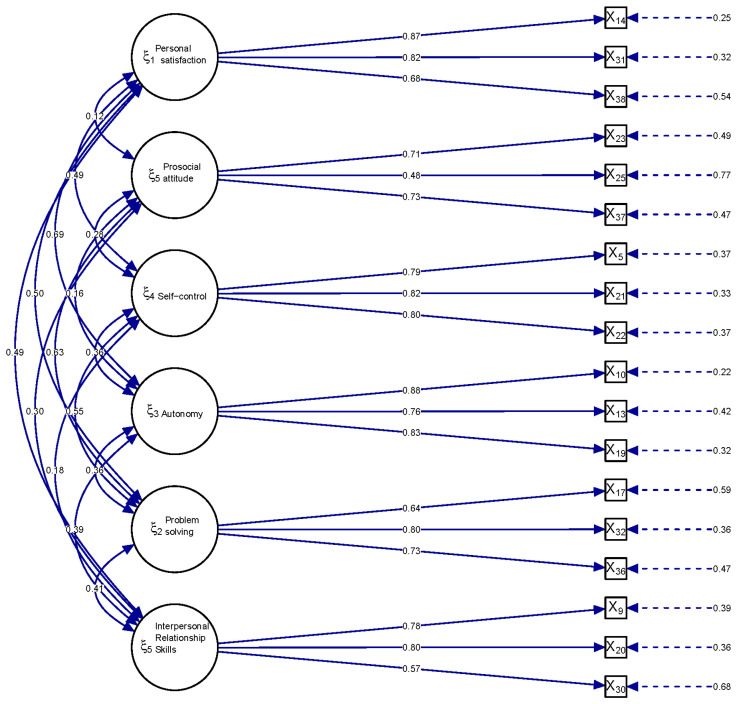
Factor loadings derived from the confirmatory factor analysis.

**Table 1 nursrep-15-00392-t001:** Sociodemographic data for the sample.

	Total (*n* = 574)	
	M	SD
Age (years-old)	20.30	5.74
	*n*	%
Gender		
Male	83	14.5%
Female	487	84.8%
Non-binary	4	0.70%
Marital status		
Single	477	83.1%
Married-partner	94	16.4%
Divorced-separated	3	0.52%
Nationality		
Spanish	550	95.8%
Foreign	24	4.18%
Employment		
Full-time	95	16.6%
Part-time	102	17.8%
Never employed	217	37.8%
Past work experience	160	27.9%
Coexistence		
Alone	5	0.87%
Parents	409	71.3%
Partner	33	5.75%
Family (others)	18	3.14%
Roommates	71	12.4%
Others	38	6.62%
Coexistence assessment		
Very good	212	36.9%
Good	259	45.1%
Neither good nor bad	82	14.3%
Poor	19	3.31%
Very poor	2	0.35%

Note. M: Mean. SD: standard deviation.

**Table 2 nursrep-15-00392-t002:** Descriptive statistics for the items.

Summary of the Content of the Items	Range	Mean	SD	Median [Q1; Q3]	Floor Effect	Ceiling Effect	Corrected Item-Total Correlation	Polychoric Correlation Item-Total Factor
Item 1: I see myself as less important than those around me	1–4	3.20	0.87	3.00 [3.00; 4.00]	5.2	44.6 ^†^	0.58	0.82
Item 2: I feel that I am someone to be trusted	1–4	3.74	0.53	4.00 [4.00; 4.00]	0.5	77.9 ^†^	0.29	0.70
Item 3: I am able to control myself when I feel negative emotions	1–4	2.71	0.91	3.00 [2.00; 3.00]	8.9	22.0 ^†^	0.43	0.85
Item 4: I worry a lot about what others think of me	1–4	2.65	0.94	3.00 [2.00; 3.00]	14.1	18.3 ^†^	0.53	0.87
Item 5: I try to improve myself as a person	1–4	3.57	0.62	4.00 [3.00; 4.00]	0.2	63.2 ^†^	0.32	.63
Item 6: I find it hard to establish deep and satisfying interpersonal relationships with some people	1–4	3.20	0.82	3.00 [3.00; 4.00]	4.0	42.0 ^†^	0.40	0.75
Item 7: I feel inept and useless	1–4	3.52	0.69	4.00 [3.00; 4.00]	1.2	62.4 ^†^	0.54	0.86
Item 8: I consider the needs of others	1–4	3.39	0.66	3.00 [3.00; 4.00]	0.5	48.6 ^†^	0.08	0.79
Item 9: I am able to control myself when I have negative thoughts.	1–4	2.68	0.89	3.00 [2.00; 3.00]	8.0	20.7 ^†^	0.46	0.83
Item 10: The opinions of others have a strong influence on me when I have to make decisions.	1–4	2.96	0.79	3.00 [2.00; 4.00]	4.0	25.6 ^†^	0.51	0.85
Item 11: I try to develop my abilities to the maximum	1–4	3.28	0.73	3.00 [3.00; 4.00]	1.6	42.5 ^†^	0.47	0.70
Item 12: I think that I am a sociable person	1–4	2.84	0.93	3.00 [2.00; 4.00]	8.4	28.2 ^†^	0.42	0.72
Item 13: I feel unsatisfied with myself	1–4	2.93	0.87	3.00 [3.00; 4.00]	8.7	25.6 ^†^	0.47	0.79
Item 14: I like to help others	1–4	3.82	0.43	4.00 [4.00; 4.00]	0.2	82.9 ^†^	0.18	0.82
Item 15: I am able to maintain a high level of self-control in conflictive situations in my life	1–4	2.91	0.82	3.00 [2.00; 4.00]	4.2	25.3 ^†^	0.46	0.84
Item 16: It troubles me when people criticize me	1–4	2.84	0.95	3.00 [2.00; 4.00]	11.1	27.0 ^†^	0.52	0.85
Item 17: When I am faced with a problem, I try to find possible solutions	1–4	3.54	0.60	4.00 [3.00; 4.00]	0.2	59.6 ^†^	0.47	0.77
Item 18: I find it hard to relate openly with my teachers/bosses	1–4	3.23	0.86	3.00 [3.00; 4.00]	5.1	46.0 ^†^	0.28	0.63

Note. SD: standard deviation. ^†^: Above the established maximum value of 15%.

**Table 3 nursrep-15-00392-t003:** Correlations between the factors of the PMHQ-SF18.

	Personal Satisfaction	Prosocial Attitude	Self-Control	Autonomy	Problem Solving	Interpersonal Relationship Skills
F1—Personal satisfaction	1					
F2—Prosocial attitude	0.07	1				
F3—Self-control	0.48	0.20	1			
F4—Autonomy	0.55	0.12	0.40	1		
F5—Problem solving	0.49	0.27	0.51	0.46	1	
F6—Interpersonal Relationship Skills	0.23	0.36	0.19	0.23	0.39	1
PMH total score	0.78	0.37	0.70	0.71	0.80	0.54

Spearman correlation.

**Table 4 nursrep-15-00392-t004:** Comparison between known groups by gender.

	Males (*n* = 83)	Females (*n* = 487)	*p*	Effect Size [IC95%]
	Mean (SD)	Mean (SD)		
F1—Personal satisfaction	9.95 (1.77)	9.61 (2.04)	0.117	0.17 [−0.06; 0.4]
F2—Prosocial attitude	10.7 (1.22)	11.0 (1.12)	0.110	−0.2 [−0.44; 0.03]
F3—Self-control	9.31 (1.97)	8.15 (2.20)	**<0.001 ***	**0.54** [0.3; 0.77]
F4—Autonomy	8.76 (2.06)	8.41 (2.33)	0.162	0.15 [−0.08; 0.39]
F5—Problem solving	10.3 (1.42)	10.4 (1.49)	0.593	−0.06 [−0.29; 0.17]
F6—Interpersonal Relationship Skills	9.19 (2.02)	9.30 (2.05)	0.669	−0.05 [−0.28; 0.18]
PMH total score	58.3 (6.17)	56.8 (7.22)	0.060	0.2 [−0.03; 0.44]

SD: Standard Deviation. * Bold: *p*-value < 0.001. |d|: effect size into the range mild-moderate (|d| > 0.50) to large-high. (|d| > 0.80).

**Table 5 nursrep-15-00392-t005:** Comparison between known groups by coexistence.

	Alone (*n* = 5)	Parents (*n* = 409)	Partner (*n* = 33)	Family (*n* = 18)	Roommates (*n* = 71)	Others (*n* = 38)	*p*	Post Hoc Comparisons (|d| [CI95%])
	Mean (SD)	Mean (SD)	Mean (SD)	Mean (SD)	Mean (SD)	Mean (SD)		
F1—Personal satisfaction	10.4 (1.14)	9.47 (2.11)	11.0 (0.92)	10.0 (1.53)	9.90 (1.77)	9.68 (1.65)	**<0.001 ***	Partner > parents (−0.76 [−1.12; −0.4]); Partner > Others (0.99 [0.49; 1.49])
F2—Prosocial attitude	11.0 (1.00)	10.9 (1.10)	11.1 (1.08)	11.1 (1.21)	11.0 (1.25)	10.9 (1.35)	0.883	
F3—Self-control	7.80 (2.77)	8.22 (2.24)	9.64 (1.93)	9.00 (2.06)	8.20 (2.07)	8.03 (1.92)	**0.008 ***	Partner > parents (−0.64 [−1; −0.28]); Partner > roommates (0.71 [0.28; 1.14]); Partner > Others (0.83 [0.34; 1.33])
F4—Autonomy	9.80 (1.79)	8.34 (2.28)	9.79 (1.54)	9.44 (2.04)	8.24 (2.54)	8.34 (2.28)	**0.003 ***	Partner > parents (−0.65 [−1.01; −0.29]); Partner > roommates (0.68 [0.25; 1.11])
F5—Problem solving	10.0 (2.35)	10.4 (1.50)	10.7 (1.35)	10.7 (1.60)	10.2 (1.44)	10.5 (1.43)	0.493	
F6—Interpersonal Relationship Skills	10.4 (1.67)	9.16 (2.01)	10.2 (2.15)	9.78 (2.02)	9.24 (2.15)	9.39 (2.13)	**0.043 ***	Partner > parents (−0.54 [−0.9; −0.18])
PMH total score	59.4 (5.46)	56.5 (7.20)	62.5 (4.67)	60.0 (6.38)	56.8 (7.19)	56.9 (6.33)	**<0.001 ***	Partner > parents (−0.86 [−1.22; −0.5]); partner > roommates (0.88 [0.45; 1.32]); Partner > others (1 [0.5;1.51])

SD: Standard Deviation. PMH: Positive Mental Health. * Bold: *p*-value <0.001. |d|: effect size into the range mild-moderate (|d| > 0.50) to large-high (|d| > 0.80).

## Data Availability

Due to ethical restrictions, the data supporting the findings of this study cannot be made publicly available.

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
