# Peer review of "Positive Mental Health Questionnaire-Short Form (PMHQ-SF18): Psychometric Properties of the Spanish Version"

_nursrep, 2025, doi:10.3390/nursrep15110392_

Round 1
Reviewer 1 Report
Comments and Suggestions for Authors
attachment

Author Response
Comments 1: Based on the text, it was difficult to determine whether the shorter Spanish version included a translation of the eighteen items from the shorter Portuguese version—or whether these eighteen items were taken from the original full Spanish version. The procedure may not be relevant here— however, back-translation is traditionally performed.
Response 1: Regarding the origin of the 18 items in the PMHQ-SF18, we would like to clarify that these items were selected from the original Spanish version of the Positive Mental Health Questionnaire (PMHQ), which contains 39 items. The selection was based on previous psychometric analyses and theoretical considerations aligned with the original conceptual model developed by one of the co-authors of this manuscript. Although the Portuguese version also underwent adaptation and item reduction, the Spanish short version was independently derived and not a direct translation of the Portuguese short version. We have now clarified this point in the revised manuscript.
As for the translation procedure, we acknowledge the importance of back-translation in cross-cultural adaptation. In this case, since the short version was derived from the original Spanish tool, back-translation was not applicable. However, we have added a note in the manuscript to explain this.
Comments 2: In their explanation, the authors refer, among other things, to the specificity of the study group, which consisted of first-year nursing students. This raises the question of their origins: were these students from Catalonia or also from other regions of Spain? Spain is a beautiful, large country with a rich and diverse culture. I wonder about the significance of the cultural context for the identified psychometric properties of the tool.
Response 2: Concerning the sample, the participants were first-year nursing students from universities in Catalonia. We agree that cultural context may influence the psychometric properties of the tool, and we have now included a reflection on this aspect in the discussion and limitation section.
Comment 3: The authors identified very interesting differences related to the gender of the students studied and their social situation – in terms of being in an (intimate?) relationship with a partner ("...our results point out that university students who are living with a partner have a higher perception of PMH (i.e., greater Personal Satisfaction, Self-Control, Autonomy, and Interpersonal Skills) than those who live with their parents or in other cohabitation situations. Does living with a partner refer to an intimate relationship or a friendship? And if in the former case it refers to an intimate relationship – how should being in an "other cohabitation situation" be understood?
Response 3: Your observations regarding the interpretation of cohabitation categories are very pertinent. In our study, “living with a partner” refers to an intimate relationship, while “other cohabitation situations” may include living with friends, roommates, or alone. We have revised the manuscript to clarify this distinction.

Reviewer 2 Report
Comments and Suggestions for Authors
This is an interesting paper!
I think that nursing students are not representative of the general population. They are more highly educated and are aware of the purpose of the study, so their responses might be skewed in some way. Also, the sample is mostly women, which does not give a good representation of the population.
Also, it would have been interesting to have asked participants' sexuality identity because there is evidence that lesbians are happier in relationships, and men are happier than women in heterosexual relationships.
Also, including variables about health and wellness would have been important, especially if this is to be used in a hospital setting.
Author Response
Comments 1: I think that nursing students are not representative of the general population. They are more highly educated and are aware of the purpose of the study, so their responses might be skewed in some way. Also, the sample is mostly women, which does not give a good representation of the population.
Response 1: We agree that nursing students are not representative of the general population. However, the aim of our study was to explore the prevalence and perceptions of dating violence within a specific university context, particularly among health sciences students. We acknowledge this limitation and have incorporated it into the discussion section, noting that the findings are not generalizable to the broader population
Response 1: It is true that the sample is predominantly female, which reflects the demographic reality of nursing programs at the participating institutions. This characteristic has also been acknowledged as a limitation in the study, and we have discussed how it may influence the results obtained.
Comments 2: Also, it would have been interesting to have asked participants' sexuality identity because there is evidence that lesbians are happier in relationships, and men are happier than women in heterosexual relationships.
Response 2: We especially appreciate this suggestion. We agree that including sexual identity would have added a valuable dimension to the analysis, as relationship dynamics can vary significantly depending on this variable. We will consider this recommendation for future research and have included it in the section on study limitations and future directions.
Comments 3: Also, including variables about health and wellness would have been important, especially if this is to be used in a hospital setting.
Response 3: Your observation about health and wellness variables is particularly relevant, especially considering the potential application of the findings in hospital settings. Although the primary focus was on positive mental health, we recognize that variables related to mental health and well-being could enrich the analysis. This suggestion has been incorporated into the section on implications for future research.

Reviewer 3 Report
Comments and Suggestions for Authors
Dear authors.
I have carefully reviewed the manuscript and found it to be a valuable contribution to the field. I would like to offer some comments and recommendations for your consideration:
Abstract
Overall, it is well written, but the terms ‘shorter’ and “shortened” are repeated several times. You could vary them slightly to improve readability with expressions such as ‘abridged version’, ‘reduced version’ or ‘compact format’.
Introduction
It is advisable to add two or three current references from 2023 to 2025 in the introduction and use them in the discussion. The objectives are clearly stated, but the research question is not explicit. It is advisable to include it in the text.
Methodology
The study has a robust methodology that is appropriate and consistent with its objectives. However, it would be advisable to include in the methodology section an explicit description of the epistemological paradigm underpinning the study (e.g., positivist or post-positivist), as well as the methodological approach (quantitative, qualitative, or mixed). This would bring greater clarity and theoretical consistency to the research design and facilitate understanding of the framework from which the validation of the instrument is approached.
Results
The presentation of the results is accurate and well argued, demonstrating an adequate interpretation of the data. Of particular note is the careful identification and analysis of items with a lower contribution, which demonstrates an in-depth and transparent analysis of the results.
Discussion
The discussion is solid, up-to-date, and coherent, highlighting the psychometric rigour and practical usefulness of the instrument. However, it shows little connection with current trends in post-COVID mental health and could be improved.
The text focuses heavily on comparisons with previous versions of the PMHQ, but could have incorporated more recent literature (2022–2025) on psychological well-being, resilience, and positive mental health in university students after the pandemic.
The discussion is highly descriptive, summarising the results rather than critically analysing them or exploring alternative explanations.
For example, the low α of F2 (prosocial attitude) is attributed solely to the nursing vocation, but other psychometric factors (e.g., reduced number of items or item wording) could have been considered.
Conclusions
The conclusion is clear and well-focused, highlighting the validity, reliability, and practical usefulness of the abbreviated instrument.
Tables and figures
The tables and figures are very well constructed, with detailed and relevant information that adequately supports the analyses performed.
References
The references are adequate but could be improved by adding some from the last three years. In terms of typography, it would be advisable to review the uniformity of style; for example, some years are in bold, and others are not.
Warm regards.
Author Response
Comments 1: Abstract
Overall, it is well written, but the terms ‘shorter’ and “shortened” are repeated several times. You could vary them slightly to improve readability with expressions such as ‘abridged version’, ‘reduced version’ or ‘compact format’.
Response 1: Thank you for your appreciation. We have reviewed the text and replaced repeated terms such as “shorter” and “shortened” with alternative expressions like “abridged version” and “compact format” to improve readability.
Comments 2: Introduction
It is advisable to add two or three current references from 2023 to 2025 in the introduction and use them in the discussion. The objectives are clearly stated, but the research question is not explicit. It is advisable to include it in the text.
Response 2: Recent references (2023–2025) have been incorporated into both the introduction and discussion sections, and the research question has been explicitly added to enhance clarity.
Comments 3: Methodology
The study has a robust methodology that is appropriate and consistent with its objectives. However, it would be advisable to include in the methodology section an explicit description of the epistemological paradigm underpinning the study (e.g., positivist or post-positivist), as well as the methodological approach (quantitative, qualitative, or mixed). This would bring greater clarity and theoretical consistency to the research design and facilitate understanding of the framework from which the validation of the instrument is approached.
Response 3: We have included an explicit description of the epistemological paradigm (post-positivist) and the methodological approach (quantitative) in the methodology section. This addition provides greater theoretical consistency and facilitates understanding of the research design.
Comments 4: Results
The presentation of the results is accurate and well argued, demonstrating an adequate interpretation of the data. Of particular note is the careful identification and analysis of items with a lower contribution, which demonstrates an in-depth and transparent analysis of the results.
Response 4: We appreciate your positive assessment of the results section. We have maintained the clear and transparent structure, emphasizing the detailed analysis of items with lower contribution.
Comments 5: Discussion
The discussion is solid, up-to-date, and coherent, highlighting the psychometric rigour and practical usefulness of the instrument. However, it shows little connection with current trends in post-COVID mental health and could be improved.
The text focuses heavily on comparisons with previous versions of the PMHQ, but could have incorporated more recent literature (2022–2025) on psychological well-being, resilience, and positive mental health in university students after the pandemic.
The discussion is highly descriptive, summarising the results rather than critically analysing them or exploring alternative explanations.
For example, the low α of F2 (prosocial attitude) is attributed solely to the nursing vocation, but other psychometric factors (e.g., reduced number of items or item wording) could have been considered.
Response 5: We agree. The discussion has been strengthened by incorporating recent literature (2022–2025) on psychological well-being, resilience, and positive mental health among university students after the pandemic. Additionally, we have added a more critical analysis, considering psychometric factors such as the reduced number of items and item wording to explain the low α in the prosocial attitude dimension.
Comments 6: Conclusions
The conclusion is clear and well-focused, highlighting the validity, reliability, and practical usefulness of the abbreviated instrument.
Response 6: The conclusion has been reviewed to maintain its clarity and emphasis on the validity, reliability, and practical usefulness of the abbreviated instrument.
Comments 7: Tables and figures
The tables and figures are very well constructed, with detailed and relevant information that adequately supports the analyses performed.
Response 7: Tables and figures have been checked to ensure uniformity and clarity, maintaining the quality highlighted in your comments.
Comments 8: References
The references are adequate but could be improved by adding some from the last three years. In terms of typography, it would be advisable to review the uniformity of style; for example, some years are in bold, and others are not.
Response 8: We greatly appreciate your thoughtful evaluation. Recent references have been added, and typographic inconsistencies have been corrected to ensure uniform formatting.
Response to the general observations: We have expanded the limitations section, noting that the sample is specific and predominantly female, which restricts generalizability. We also acknowledge that data on health and mental well-being were not collected, which could have enriched the interpretation of the results. For future research, we propose exploring the validity of the instrument in more diverse Spanish-speaking populations and across different cultural contexts, as well as including sexual identity as a relevant variable to better understand relationship dynamics

Round 2
Reviewer 3 Report
Comments and Suggestions for Authors
Dear authors.
Your manuscript has been reviewed, and I have noticed a significant improvement. My recommendations below focus on the English language for clarity and consistency.
Comments on the Quality of English Language
Although the manuscript is well written I recommend reviewing the English to further enhance clarity. In particular, I noticed a mixture of American and British spelling (for example, analyzed on line 168 vs analyse on line 104), so it would be best to standardize the manuscript to a single style. Additionally, there are some errors such as scores distribution on line 160; the plural should not be used when a noun is acting as an adjective.
Overall, your work shows great progress, and I appreciate your effort and dedication.
Warm regards.
The reviewer
Author Response
Comments 1: Although the manuscript is well written I recommend reviewing the English to further enhance clarity.
Response 1:We have conducted a thorough language revision with the support of a professional translator to ensure clarity, coherence, and consistency throughout the text.
Comments 2: In particular, I noticed a mixture of American and British spelling (for example, analyzed on line 168 vs analyse on line 104), so it would be best to standardize the manuscript to a single style.
Response 2: We standardized the spelling to British English across the manuscript to avoid inconsistencies such as "analyzed" vs. "analyse".
Comments 3: Additionally, there are some errors such as scores distribution on line 160; the plural should not be used when a noun is acting as an adjective.
Response 3: We corrected grammatical issues, including the phrase "scores distribution" on line 160, which now reads "score distribution" to ensure proper noun usage.
